# Nicotinamide Riboside, a Promising Vitamin B_3_ Derivative for Healthy Aging and Longevity: Current Research and Perspectives

**DOI:** 10.3390/molecules28166078

**Published:** 2023-08-15

**Authors:** Andrei Biţă, Ion Romulus Scorei, Maria Viorica Ciocîlteu, Oana Elena Nicolaescu, Andreea Silvia Pîrvu, Ludovic Everard Bejenaru, Gabriela Rău, Cornelia Bejenaru, Antonia Radu, Johny Neamţu, George Dan Mogoşanu, Steven A. Benner

**Affiliations:** 1Department of Pharmacognosy & Phytotherapy, Faculty of Pharmacy, University of Medicine and Pharmacy of Craiova, 2 Petru Rareş Street, 200349 Craiova, Dolj County, Romania; andreibita@gmail.com (A.B.); ludovic.bejenaru@umfcv.ro (L.E.B.); george.mogosanu@umfcv.ro (G.D.M.); 2Department of Biochemistry, BioBoron Research Institute, S.C. Natural Research S.R.L., 31B Dunării Street, 207465 Podari, Dolj County, Romania; mariaviorica.bubulica@gmail.com (M.V.C.); gabriela.rau@umfcv.ro (G.R.); johny.neamtu@umfcv.ro (J.N.); 3Department of Analytical Chemistry, Faculty of Pharmacy, University of Medicine and Pharmacy of Craiova, 2 Petru Rareş Street, 200349 Craiova, Dolj County, Romania; 4Department of Pharmaceutical Technology, Faculty of Pharmacy, University of Medicine and Pharmacy of Craiova, 2 Petru Rareş Street, 200349 Craiova, Dolj County, Romania; oana.nicolaescu@umfcv.ro; 5Department of Biochemistry, Faculty of Medicine, University of Medicine and Pharmacy of Craiova, 2 Petru Rareş Street, 200349 Craiova, Dolj County, Romania; andreeaneamtu1989@yahoo.com; 6Department of Organic Chemistry, Faculty of Pharmacy, University of Medicine and Pharmacy of Craiova, 2 Petru Rareş Street, 200349 Craiova, Dolj County, Romania; 7Department of Pharmaceutical Botany, Faculty of Pharmacy, University of Medicine and Pharmacy of Craiova, 2 Petru Rareş Street, 200349 Craiova, Dolj County, Romania; cornelia.bejenaru@umfcv.ro (C.B.); antonia.radu@umfcv.ro (A.R.); 8Department of Physics, Faculty of Pharmacy, University of Medicine and Pharmacy of Craiova, 2 Petru Rareş Street, 200349 Craiova, Dolj County, Romania; 9Foundation for Applied Molecular Evolution (FfAME), 13709 Progress Avenue, Room N112, Alachua, FL 32615, USA; sbenner@ffame.org

**Keywords:** nicotinamide riboside, vitamin B_3_ derivative, efficacy, safety, healthy aging, longevity

## Abstract

Many studies have suggested that the oxidized form of nicotinamide adenine dinucleotide (NAD^+^) is involved in an extensive spectrum of human pathologies, including neurodegenerative disorders, cardiomyopathy, obesity, and diabetes. Further, healthy aging and longevity appear to be closely related to NAD^+^ and its related metabolites, including nicotinamide riboside (NR) and nicotinamide mononucleotide (NMN). As a dietary supplement, NR appears to be well tolerated, having better pharmacodynamics and greater potency. Unfortunately, NR is a reactive molecule, often unstable during its manufacturing, transport, and storage. Recently, work related to prebiotic chemistry discovered that NR borate is considerably more stable than NR itself. However, immediately upon consumption, the borate dissociates from the NR borate and is lost in the body through dilution and binding to other species, notably carbohydrates such as fructose and glucose. The NR left behind is expected to behave pharmacologically in ways identical to NR itself. This review provides a comprehensive summary (through Q1 of 2023) of the literature that makes the case for the consumption of NR as a dietary supplement. It then summarizes the challenges of delivering quality NR to consumers using standard synthesis, manufacture, shipping, and storage approaches. It concludes by outlining the advantages of NR borate in these processes.

## 1. Introduction

Nicotinamide riboside (NR) is widely used as a dietary supplement. Structurally, it is a form of vitamin B_3_ (nicotinic acid, niacin, NA), incorporating into its structure more elements of nicotinamide adenine dinucleotide (in its oxidized form, NAD^+^) [1]. NR influences, in particular, energy metabolism and neuroprotection [2,3,4].

From a non-medical perspective, as a part of the NAD^+^ cofactor, NR is also inferred to be a vestige of the “ribonucleic acid (RNA) world” [5], an episode of life on early Earth where RNA was the only encoded component of both metabolism and genetics. This inference is supported by the ubiquity of NAD^+^ in all branches of the tree of modern life [6].

NR is, however, a rather reactive molecule. Its glycosidic bond joins a positively charged pyridinium heterocycle to a carbohydrate. This bond is therefore especially unstable to cleavage, making NR difficult to synthesize, store, and transport. This creates broad utility for any method to synthesize NR, as well as broad utility for derivatives of NR that are more stable.

Stable forms of NR are especially important today. In July 2013, NR became accessible in dietary supplements in the form of NR chloride (NRCl). This is sold and widely consumed as a dietary supplement under various trademarks, including Tru Niagen™ and Niagen^®^ (produced by ChromaDex, Los Angeles, CA, USA). Other products containing NRCl have been commercially available [7,8]. Interestingly, analysis of some of these products at various times has shown that they do not contain pure NR but also products undoubtedly arising from the reactivity of NR.

Notwithstanding, commercial NR products are suggested to elevate the level of NAD^+^ in those who consume it [8]. Studies report that chronic NR supplementation (NRS) is well tolerated and elevates NAD^+^ in healthy middle-aged and older adults [2,3]. Other studies suggest NR oral bioavailability in mice and humans [9]. NR as a dietary supplement has also been the target of clinical studies [2,3]. These have proven its effect of boosting NAD^+^ in its consumers [8,10]. These facts make it timely to review the current state of the art.

NR, a pyridine-nucleoside form of vitamin B_3_, consists of nicotinamide (NAM) and ribose as its fundamental components. It is found in various sources, such as milk, yeast, beer, bacteria, and mammals. NR-enriched foodstuffs per se are not well established. Presumably, products that contain yeast are excellent natural sources for the compound [11,12]. However, dairy products have also been noted to contain NR [9,13]. In general, the amounts of NR in foodstuffs are quite low, most likely at micromolar levels.

NR shares biological properties with other NAD^+^ precursors, specifically NAM and NA. Indeed, these are old or “classical” vitamins B_3_. Unlike NR, NAM and NA have disadvantages when consumed. Hepatotoxicity can be side effect associated with NAM, whereas a recent preclinical study indicates that NAM remains in the rat body for a shorter duration when compared to NR. NA taken in large amounts is associated with adverse effects, including cutaneous flushing when administered in an immediate release formulation. Sustained release formulations may cause hepatotoxicity.

A precursor that is structurally more advanced toward NAD^+^ is nicotinamide mononucleotide (NMN). NMN has been recently discovered to be converted extracellularly to NR, which is transported into cells [11,14,15]. Since NMN is converted to NR in the body and the price of NMN is half of the price of NR, technology producing NR to replace NMN as a dietary supplement is especially needed [16]. Among the NAD^+^ precursors, NR may be preferred, as it produces fewer reported unfavorable side effects [7].

In its classical metabolic roles, NAD^+^ is a versatile receiver of hydride equivalents, becoming NADH. Analogous chemistry is seen with its phosphorylated derivative, nicotinamide adenine dinucleotide phosphate (NADP^+^ to NADPH). Traditionally, NAD^+^ and its derivatives function as coenzymes for dehydrogenases and oxidoreductases, playing crucial roles in fundamental energy metabolism processes, such as glycolysis, the citric acid cycle, and mitochondrial (MT) electron transport. In addition to its classical functions, NAD^+^ also serves as a vital substrate for signaling enzymes, e.g., poly [adenosine diphosphate (ADP)-ribosyl] polymerases, sirtuins (SIRTs), and ADP-ribosyl-transferases. These enzymes are referred to as “NAD^+^ consumers” [17,18,19,20,21].

As is classically understood, as well as from recent discoveries, NAD^+^ is seen as a crucial and abundant metabolite present in all mammalian cells. It participates in a wide array of cellular mechanisms, including essential processes such as metabolism and cell signaling that are vital for survival. This notwithstanding, the limited presence of NR in food sources (with some quantitative studies available), together with the challenges associated with obtaining significant quantities of pure NR as a dietary supplement, have hindered research on the effects of NR on cells and tissues [22,23]. Nevertheless, recently, synthesis techniques for producing NR have been notably advanced, resulting in new and refined approaches [24]. However, these new technologies were developed for cell-based investigations and animal feeding tests, not for human consumption [25,26].

Non-classical roles also exist for NAD^+^. For example, NAD^+^ activates SIRTs and supports the MT response to unfolded proteins. Pursuing these non-classical roles, NAD^+^ metabolism is seen to be involved in an extensive spectrum of pathologies, including cancer, neurodegenerative disorders, cardiomyopathy, obesity, diabetes, and even hearing loss [4,27,28,29,30,31].

In many tissues (e.g., brain, muscle, skin, liver, pancreas, and adipose tissue), the level of NAD^+^ decreases with age. A separate decrease in the NAD^+^/NADH ratio affects the cellular redox state, highlighted by anaerobic glycolysis and oxidative phosphorylation (OXPHOS). This decreases, it is thought, the capacity of cells to generate adenosine triphosphate (ATP) [32]. Consequently, healthy aging and longevity are thought to be related to NAD^+^ metabolism, mainly through NR and NMN, two essential and well-studied NAD^+^ derivatives. One argument for NAM-related dietary supplements is that these can have prophylactic and therapeutic impact on functional decline, improving age-associated neurodegenerative, cardiovascular (CV), and metabolic diseases and conditions, and promoting the beneficial effects of calorie restriction (CR) [4,33,34,35,36].

As a precursor of NAD^+^, NR is also proposed to be important in regulating oxidative stress (OXS), inhibiting oxidative injury and inflammatory response, with beneficial effects in the treatment of sepsis [37], systemic lupus erythematosus (SLE) [38], and necrotizing enterocolitis (NEC) [39]; furthermore, NR showed significant protective effect on lung injury caused by paraquat (PQ) in mice [40].

Separately, sepsis-caused multiple organ failure is the major risk of morbidity and mortality in intensive care units. In experimental models of mouse sepsis induced by lipopolysaccharide (LPS) injection, feces injection in peritoneum, or by cecal ligation and puncture, NR inhibited plasma high mobility group box 1 (HMGB1) release, OXS, and tissue infiltration, increased endogenous antioxidant ability, prevented lung and heart injury, and improved survival [37,41]. This is mediated via NAD^+^/SIRT1 signaling.

NR could be a potential adjuvant for SLE treatment. In vitro testing in monocytes from patients with SLE showed that NR restricted autophagy (ATG) and attenuated interferon-beta (IFN-β) release in an NAD^+^-dependent manner, but also through inosine signaling [38]. Also, in the mouse experimental NEC model, NR administration alleviated OXS, increased NAD^+^ levels and intestinal microcirculatory perfusion, and relieved signs of endothelial dysfunction by modulating the SIRT1-associated endothelial nitric oxide synthase (eNOS) acetylation/deacetylation pathway [39,42].

Compared with the control (CON) group, which received saline by one-time gavage, intraperitoneal (i.p.) injection of 300 mg/kg NR solution led to the inhibition of the inflammatory response, peroxidation injury, and apoptosis at the lung level and to the survival time prolongation of the PQ intoxicated mice, mainly by upregulating SIRT1 and nuclear factor erythroid 2 (NF-E2)-related factor 2 (NRF2) protein expression [40].

We recently reported an inexpensive synthesis of borate-stabilized NR based on models for how nucleosides might have been formed on a prebiotic Earth. Unlike many commercially available NR products, borate-stabilized NR can be delivered in pure form, where it exhibits high stability against thermal and chemical decomposition. Thus, NR borate is a promising nutraceutical to replace NRCl, as currently sold on the United States (U.S.) market. This paper reviews the current research supporting the value of NR for healthy aging and longevity, the perspectives of senotherapeutic NAD^+^ supplementation, and thus forms of NR that can be made, shipped, and stored without decomposition.

## 2. Effects of Nicotinamide Riboside on Several Organs and Systems

### 2.1. Nicotinamide Riboside and the Nervous System

The effects of NR on energy metabolism and neuroprotection were highlighted with the first research about this compound. In yeast and mammals, NR is metabolized by two main pathways: (i) degradative processes from which NAM results; (ii) synthetic processes exploiting NR kinases (NRK1, NRK2), resulting in increased tissue NAD^+^. This increase is reported to increase insulin sensitivity, MT biogenesis, and SIRT functions.

Thus, in experimental models of Alzheimer’s disease (AD), NR in larger amounts than is found naturally in food leads to brain-protective effects by the stimulation of NAD^+^ anabolism [7]. NR is a neuroprotective factor that improves cognition after brain injuries, such as acute ischemia. In an experimental model, 300 mg/kg NRCl was i.p. administered, 20 min after reperfusion, in mice with middle cerebral artery (MCA) occlusion. The NRCl-treated group exhibited better memory function and recovery of learning in the Morris water maze test.

Following acute treatment with NRCl, apoptosis in the hippocampus, neuronal loss, and hippocampal infarct volume were decreased. NRCl also increased the measured amounts of NAD^+^ and ATP, and stimulated adenosine 5′-monophosphate (AMP)-activated protein kinase and SIRT1, as observed by high-performance liquid chromatography (HPLC) and Western blot assays [43]. Moreover, the clinical phenotype and T-cell survival/function were improved by S-adenosylmethionine (SAM) and NR co-administration for patients with phosphoribosyl pyrophosphate synthetase 1 (PRPS1) deficiency (Arts syndrome, manifested by serious neurological and immunological deficiencies especially in males) [44].

NR is reported to be beneficial in other neurological syndromes. For example, Gulf War illness (GWI) is a chronic neuropsychiatric disorder characterized by neurocognitive deficits arising from OXS, neuroinflammation, and neuronal damage. Currently, no effective treatment of GWI is known [45]. However, in a GWI experimental model, NR-mediated SIRT1 activation restored brain MT bioenergetics dysfunction (following astroglia activation) and reduced neuroinflammation. Here, NR was administered as a dietary supplement, 100 μg/kg daily for two months. In the brain of NR-treated GWI mice, a decrease in lipid peroxidation and proinflammatory cytokines was closely correlated to the increase in deacetylation of the peroxisome proliferator-activated receptor gamma coactivator-1 alpha (PGC-1α) and nuclear factor-kappaB (NF-κB) p65 subunit [46].

Many central nervous system (CNS) disorders arise from dysfunctionalities in the gut microbiota–brain axis. In the mice brain, NRS provided protection against alcohol-induced depressive behavior and decreased the level of anti-inflammatory (interleukin (IL)-10 and transforming growth factor (TGF)-β) and proinflammatory (IL-1β, IL-6, and tumor necrosis factor (TNF)-α) cytokines. In the hippocampus, NR significantly reduced the number of activated microglia and the inhibition of the protein kinase B (AKT)/glycogen synthase kinase 3 beta (GSK3β)/β-catenin signaling pathway and increased the brain-derived neurotrophic factor (BDNF). Similar with donor mice, in recipient mice, fecal microbiota transplantation (FMT) improved the microglial activity, the level of BDNF and cytokines, the activation of AKT/GSK3β/β-catenin signaling pathway, and cognitive behavior [47].

Intracortical administration of NR protects against excitotoxicity-induced axonal degeneration (AxD) and decreases the brain damage provoked by injection of *N*-methyl-D-aspartate (NMDA). Both NR and NAD^+^ prevented neuronal death due to the axonal stress. However, NR exhibited better neuroprotection than NAD^+^ at the level of cortical neurons [48].

Moreover, in rats with TNF-induced optic nerve degeneration, intravitreal injection of NR showed significant axonal protection. In retina and optic nerve, NR upregulated the levels of SIRT1 ATG pathway, decreased the p62 protein levels and increased the levels of microtubule-associated protein 1A/1B-light chain 3-II (LC3-II) that localizes with the MT inner membrane. The presence of NRK1, upregulated by NR administration, was also evidenced in the retinal and optic nerve fibers and in the retinal ganglion cells (RGCs) [49,50].

This research continues. For example, Sun et al. (2023) recently showed that NAD^+^ depletion mediates cytotoxicity in human neurons with ATG deficiency, leading to cytotoxicity and neurodegeneration. The research was conducted on ATG-deficient (ATG5^−/−^) human embryonic stem cells (hESCs) representing a “human neuronal platform”. In ATG5^−/−^ neurons, NR treatment improved cell viability by the restoration of MT bioenergetics and proteostasis [51].

In experimental (preclinical) models and also in clinical studies, NR improved ataxia scores and immunoglobulin G (IgG) levels in ataxia telangiectasia (A–T). In ATM-deficient mice, which is a model for the A–T phenotype in humans, NR administration improved motor function and prevented neuroinflammation, neurodegeneration, MT dysfunction and cellular senescence [52]. NR dietary supplementation was well tolerated by A–T patients; ataxia, dysarthria, and the quality of life (QoL) were all reportedly improved without adverse effects [53,54].

In cluster of differentiation (CD) 157 knockout (KO) male mice, oral administration of NR increased the levels of oxytocin (OT) in cerebrospinal fluid (CSF), stimulating the release of this anxiolytic factor during stress, and corrected the fearful and anxiety-like behaviors and the social deficits [55]. A recent study highlighted the remission of social behavior impairment by oral gavage administration of NR for 12 days in CD157, but not in CD38 KO mice [56].

In cell culture (in vitro) and in mouse models of amyotrophic lateral sclerosis (ALS), NR and pterostilbene (PT) supplementation cooperatively delayed motor neuron failure, decreased the levels of neuroinflammation markers in the spinal cord, influenced the muscle metabolism, and increased to a small extent the survival of hSOD1^G93A^ transgenic mice [57]. Further, in SOD1^G93A^ transgenic and wild-type (WT) mice, starting at 50 days of age, oral administration of NR (20 mg/mL) improved MT proteostasis and adult neurogenesis. In the brain of SOD1^G93A^ mice, neural stem cells (NSCs)/neuronal precursor cells (NPCs) proliferation and migration were enhanced following the NRS [58]. A recent clinical trial evidenced that NR and PT co-administration proved to be effective in inhibiting OXS-induced cellular damage, the major pathophysiological mechanism of ALS [59,60].

In mouse models for Cockayne syndrome (CS) (CSA^−/−^ and CSB^m/m^, modeling human CS), NR is reported to protect from noise-induced hearing loss (NIHL). The main mechanism of action involves the activation of NAD^+^/SIRT3 pathway, which contributes to the reduction in spiral ganglia neurite degeneration caused by intensive noise exposure (NE). NAD^+^ supplementation could prove beneficial for the treatment of CS-related hearing loss at a young age and age-related hearing loss (ARHL) affecting elderly individuals [61,62,63].

In Sprague–Dawley female rats, oral administration of NR (200 mg/kg) for four weeks relieved the nociceptive and aversive dimensions of intravenous (i.v.) paclitaxel (PTX)-induced peripheral neuropathy. In addition, NRS increased NAD^+^ plasma level by 50% and did not influence the myelosuppressive properties of PTX, the adverse locomotor effects not being observed [64].

In male rats, NR oral treatment (500 mg/kg) alleviated PTX-induced corneal and somatic hypersensitivity to tactile stimuli, with no suppression of basal tear production/chemosensitivity and without altering the corneal afferent density. NR also reversed the PTX-induced hindpaw hypersensitivity to cool and tactile stimuli, with no inverting of the non-peptidergic intraepidermal nerve fiber (IENF) loss [65].

In the recent studies regarding neuroprotective therapy for glaucoma, in a mouse model of acute RGC damage caused by optic nerve crush (ONC), and with chronic RGC degeneration obtained by intracameral injection of microbeads that induced ocular hypertension, orally administered NR removed retinal inflammation, as shown by immunofluorescence staining for glial fibrillary acidic protein (GFAP). It also enhanced survival and preserved the function of RGC [66,67]. In a mouse model of light-induced retinal degeneration, i.p. injected NR increased the NAD^+^ level and had a protective effect on retinal function through the restoration of the photoreceptor cell layers, by the reduction in the inflammation and by the limitation of the consequences of apoptosis process [68].

In 125 patients with primary open-angle glaucoma (POAG), a randomized double-blind, placebo (PLA)-controlled trial showed an effect of NR (300 mg for 24 months) in slowing down optic nerve degeneration. Compared with PLA, NR patients had a lower degree of progressive retinal nerve fiber layer (RNFL) thinning and visual field (VF) loss [69,70].

The protective effects of NR were also investigated on hydrogen peroxide-induced oxidative damage in lens epithelial cells. In vitro, on SRA01/04 cell line, NR significantly reduced apoptosis and the generation of reactive oxygen species (ROS), increased cell viability, and improved levels of superoxide dismutase (SOD), catalase (CAT), total glutathione (GSH), and MT membrane potential. NR was found to reduce OXS damage through the targeting of mitogen-activated protein kinase (MAPK) and Janus kinase 2 (JAK2)/signal transducer and activator of transcription 3 (STAT3) pathways [71].

Pang et al. (2021) reported that some aberrant metabolic pathways during pregnancy, such as NAD^+^, OXPHOS, tricarboxylic acid (TCA) cycle and tryptophan metabolism, underlie Zika virus (ZIKV)-induced microcephaly in newborns. Preclinical experiments suggested that in ZIKV-infected mice, NRS improved survival, reduced the apoptosis, and increased the thickness of the cerebral cortex [72].

### 2.2. Nicotinamide Riboside and the Cardiovascular System

Dietary supplementation by NAD^+^ precursors (e.g., NR) appears to have a useful impact in many areas of cardiovascular diseases (CVDs). Here, the goal is to improve overall cardiometabolic health by increasing OXPHOS capacity and mitophagy, removing mitochondrial deoxyribonucleic acid (mtDNA) mutations, and for cardioprotective effects in ischemia–reperfusion (I–R) injury, arrhythmias, heart failure (HF), myocardial infarction (MI), and high blood pressure [73,74,75,76].

For example, in an experimental model of cafeteria (CAF) diet-induced obesity (DIO), oral administration of 400 mg/kg of NR combined with CR (−62% kcal) for 28 days decreased the weight of obese male Wistar rats, their visceral and subcutaneous adiposity, their triglyceride (TG)/high-density lipoprotein (HDL) ratio, and their heart size. Also, insulin resistance and antioxidant capacity (glutathione peroxidase (GPx) and CAT enzymatic markers of cardiac OXS) were improved by NRS [77]. In a mouse model, NR treatment for 30 days led to the NAD^+^/NADH ratio balancing, lowering acetylation level, improving MT function and HF with preserved ejection fraction (EF) phenotypes [78]. In a murine model of myocardial hypertrophy induced by transverse aortic constriction (TAC) surgery, NR dietary supplementation reduced levels of inflammatory cytokines (IL-1β, TNF-α), mitigated the NLR family pyrin domain containing 3 (NLRP3) inflammasome activation, the elevation of myocardial NAD^+^ level, and the improving of cardiac dysfunctions and morpho-functional changes (myocardial hypertrophy). The research led to the observation that the NAD^+^/SIRT3/manganese superoxide dismutase (MnSOD) signaling pathway was also regulated by NRS [79]. Moreover, in mice, NR administration inhibited TAC-induced endothelial-to-mesenchymal transition of endothelial cells (ECs) and promoted MT unfolded protein response leading to improvement in prohibiting proteins’ expression and to the decrease in cardiac fibrosis (CF) progression [80].

Recent clinical trials highlighted the cardioprotective role of oral NRS (usually 1 g twice daily, a well-tolerated dose) for ischemic heart disease patients diagnosed with atrial fibrillation (AF) [81], HF patients with MT dysfunction and peripheral blood mononuclear cells (PBMCs) inflammatory activation [82], or clinically stable HF patients with reduced EF [83].

In murine models, boosting NAD^+^ levels using dietary NRS appeared to reduce the development of aortic aneurysms and sudden death by aortic ruptures. Acute aortic aneurysms and lethal ruptures were induced by angiotensin II (Ang II) administration to apolipoprotein E (ApoE)-deficient mice fed throughout the entire experiment with a Western diet (WD). Fatal aortic ruptures caused by atherosclerosis and hypertension are reported to be prevented by boosting MT respiration [84].

Additional studies in this space have accumulated. Thus, NRK2 is reported to limit dilated cardiomyopathy (DCM) in mice with chronic pressure overload (PO). In addition, Ang II-induced cardiomyocyte death is reported to be mitigated by NRK2 overexpression [85]. NR administration preserves cardiac function in a mouse model of DCM and TAC-induced cardiac hypertrophy, alleviating HF development by stabilizing NAD^+^ levels in the myocardial tissue.

NA adenine dinucleotide, methylnicotinamide (MeNAM), and *N*^1^-methyl-4-pyridone-5-carboxamide represent the three biomarkers with increased myocardial levels due to NRS [86]. In murine models, NR was i.v. administered to assess its ability to mitigate doxorubicin (DOX)-induced cardiomyopathy. OXS, autolysosomes accumulation and autophagic flux blockade because of DOX cardiotoxicity were all prevented by NR via NAD^+^/SIRT1 signaling pathway [87,88]. For the specific pathogen-free male Wistar rats, the protective effect exhibited significant levels through the preventive use of NR [89].

NR also attenuated cardiac and I–R injury. In the ischemic heart, NRK2 regulates metabolic adaptation and MT function post-MI [90]. In a cardiac I–R injury experimental (mouse) model, orally administered NR and resveratrol (RSV) nanocrystal self-assembled microspheres (NR/RSVms) for eight hours decreased MI, with no important adverse effects on internal organs [91]. Also, in a fentanyl–midazolam anesthetized I–R injury rat model, NR was i.v. administered as a bolus before the ischemia started. It was found that NR could target cardiac I–R injury as a promising cardioprotective natural compound [92]. Moreover, in a mouse model of superior mesenteric artery ischemia, NRS improved microcirculation and mesenteric vessels relaxation and protected the intestinal wall against I–R injury [93].

In human aortic endothelial cells (HAECs) and in murine aortic rings, NR acts as a vasoprotective agent at the endothelial level by modulation of intracellular NAD^+^ and inhibition of inflammation, as evidenced by the decreased expression of von Willebrand factor (vWF) and intercellular adhesion molecule-1 (ICAM-1) [94].

### 2.3. Nicotinamide Riboside and the Digestive System

In mice subjected to partial hepatectomy, boosting NAD^+^ by putting NRS in the drinking water promoted liver regeneration. This was interpreted as increasing DNA biosynthesis at the hepatocyte level and amelioration of hepatic steatosis (HS) [95]. In a high-fat diet (HFD)-fed murine model, NR treatment enhanced hepatic MT function and NAD^+^ levels and prevented lipid accumulation in the liver [96]. In vitro, HS was induced in AML12 mouse hepatocytes treated with 250 μM palmitic acid for 48 h. Cell exposure to NR (10 μM and 10 mM) for 24 h did not affect morphology nor viability; instead, it was observed a decrease in TNF-α and IL-6 proinflammatory markers and an increase in *SIRT1* gene activity, PGC-1α, transcription factor A, carnitine palmitoyltransferase 1, uncoupling protein 2, mtDNA and MT biogenesis [97].

Supplementation with NAD^+^ precursors (NR, NMN) can prevent liver injury. Proposals to treat non-alcoholic fatty liver disease (NAFLD) NRK1 by targeting HS are based on the discovery that in aged mice or in mice treated with HFD, NRK1 overexpressed by adenovirus-mediated gene transduction regulates lipid metabolism (mainly TG level) and NAD^+^ biosynthesis in the liver, insulin sensitivity, and glucose tolerance [98]. In aged mice with a moderate NAFLD-like phenotype, NRS (2.5 g/kg food for three months) significantly reduced the levels of total cholesterol (TC), TG, aspartate aminotransferase (AST), and alanine aminotransferase (ALT) and increased the amount of liver NAD^+^, diminished inflammatory infiltration, and mitigated HS and liver fibrosis (LF) [99,100,101]. In a diet-induced NAFLD mice model, supplementation with a multi-ingredient mixture of betaine, *N*-acetylcysteine, L-carnitine, and NR slowed NAFLD progression and influenced the gut–liver axis through the correction of intestinal microbiota dysbiosis and modulation of short-chain fatty acids (SCFAs) levels in feces [102].

Daily NR and PT (NRPT, Basis™) co-administration in 111 adults with NAFLD was assessed in a six-month prospective, randomized, double-blind, PLA-controlled clinical trial. Here, NRPT treatment was well tolerated, and significantly decreased serum levels of ceramide 14:0 (a toxic lipid), ALT, and gamma-glutamyltransferase (GGT) [103].

In an HFD-induced male C57BL/6J mice model of LF, NR administration (400 mg/kg/day for 20 weeks) attenuated the development of LF. The body weight, the amount of collagen in the liver, and the activation of hepatic stellate cells (HepSCs) were significantly reduced by NRS, compared to liver inflammation, HS, and ALT levels, which have not been mitigated [104]. In a similar context, NR exhibited a protective effect against LF induced by carbon tetrachloride in mice through an increase in *SIRT1* gene activity, modulation of Smads signaling pathway acetylation, and suppression of TGF-β-induced HepSCs activation [105].

NR treatment (400 mg/kg for 16 days) attenuated liver injuries induced by alcohol in C57BL/6J mice fed a Lieber–DeCarli ethanol liquid diet. The protective effect of NRS involves SIRT1 activation, increasing hepatic NAD^+^ levels and PGC-1α deacetylation, enhancing MT biogenesis/functionality, and decreasing OXS status and ethanol-induced lipid accumulation [106].

A recent in vitro study in a human hepatic (HepG2) cell line model of citrin deficiency highlighted the potential use of NRS for balancing dysregulated glycolysis and fatty acid β-oxidation [107].

NRS has even been suggested as being useful to manage the sequelae of infectious diseases. For example, as part of research aimed at new treatments for severe acute respiratory syndrome coronavirus 2 (SARS-CoV-2) disease, it was highlighted that NRS significantly inhibited the murine hepatitis virus (MHV) replication by increasing the level of NAD^+^, activation of the TCA cycle and of MT metabolism [108].

### 2.4. Nicotinamide Riboside and the Urinary System

In the U.S. hospitals, about 3–10% of the adults suffer from acute kidney injury (AKI). NAD^+^ supplementation, by oral administration of NAM, NA, NR, dihydronicotinamide riboside (reduced NR, NRH) and NMN, and SIRT activation may be useful in the prophylaxis and treatment of AKI, if some recent studies on MT metabolism are developed [109,110,111,112,113]. For example, in an experimental model of newborn (one day) mice from mothers fed a low protein diet, NRS restored the SIRT3 expression, by the induction of PGC-1α, improved MT and cellular protection against OXS, increasing the density of nephrons, renal capillaries, and glomerular podocytes [114].

The effect of four NRPT doses (NR/PT—1.25 g/50 mg, 2.5 g/100 mg, 3.75 g/150 mg, 4 g/200 mg), administered twice a day for two days, was tested in a randomized, double-blind, PLA-controlled study in 24 hospitalized AKI patients. Serum NAD^+^ levels were increased by NRPT treatment. Safety was assessed by various assays, such as estimated glomerular filtration rate (eGFR), creatinine levels, electrolyte levels, hepatic function parameters, and blood count; these all remained unchanged. Minor gastrointestinal (GI) side effects were reported only by three patients. NRPT supplementation up to 1 g/200 mg twice a day was reported as being safe and well tolerated in hospitalized AKI patients [110].

In another recent preclinical study, in rats, NR prophylactic supplementation did not mitigate tubular impairment and induction of profibrotic genes neither in bilateral I–R injury-induced long-term AKI nor in chronic kidney disease (CKD) experimental model, even though an increased NAD^+^ blood levels were evidenced [115]. However, Doke et al. (2023), in a study in male mice with cisplatin (CPT)- or I–R-induced AKI, suggested that NR and NMN supplementation restored NAD^+^ serum levels and improved MT metabolism and kidney functionality [116].

A randomized PLA-controlled, double-blind, crossover trial was conducted for six weeks to determine the impact of NR (1 g/day) and coenzyme Q10 (CoQ10, 1.2 g/day) on exercise tolerance and metabolic profile in 25 CKD patients with eGFR of <60 mL/min/1.73 m^2^. Compared with PLA, only systemic MT metabolism and lipid profile (e.g., TG, free fatty acids, ceramides) were improved by NR and CoQ10 co-administration, but not aerobic capacity (VO_2_ peak) or total work efficiency [117].

### 2.5. Nicotinamide Riboside and the Musculoskeletal System

Numerous preclinical studies have suggested that NR and NMN dietary supplements increase NAD^+^ levels in the skeletal muscle and help protect musculoskeletal system from age-related metabolic dysfunction. NAD^+^ biosynthesis pathways/homeostasis and its bioavailability in skeletal muscle cells are influenced by NAM phosphoribosyltransferase (NAMPT), NRK1, and NRK2 enzymes [118,119]. After the application of NRS in hexose-6-phosphate dehydrogenase KO (H6PDKO) mice, NAD^+^/NADH ratio was elevated but without any effect on insulin sensitivity, MT respiratory dysfunctions (MT acylcarnitine), acetyl coenzyme A metabolism and endoplasmic reticulum OXS [120,121].

In a double-blind, PLA-controlled, randomized, crossover trial for 21 days, 12 aged men were supplemented with 1 g/day NR. It was found that both MT bioenergetics and skeletal muscle NAD^+^ metabolism were stimulated by NRS; at the same time, the amounts of serum inflammatory cytokines were reduced [122]. Moreover, in a randomized, double-blinded, PLA-controlled, crossover intervention study in 13 healthy obese humans, NRS (1 g daily for six weeks) induced some minor modifications in body composition and sleeping metabolic rate, increased the level of NAD^+^ metabolites, such as NA adenine dinucleotide and MeNAM, and altered the acylcarnitine concentrations in the skeletal muscle [123].

Recent studies showed that, in healthy humans, oral NAD^+^ supplementation (NR, NMN) does not alter whole-body or skeletal muscle metabolic responses to an endurance exercise [124,125]. During endurance exercises, one week of NRS (1 g daily) for eight young males did influence neither skeletal muscle MT respiration nor auto-PARylation of poly ADP-ribose polymerase 1 (PARP1), acetylation of tumor protein 53 (p53) Lys382 and MnSODLys122, or overall acetylation [126]. A recent randomized controlled trial testing of in-home aerobic and exercise training and 16 weeks NRS targeted the skeletal muscle MT OXPHOS and improved the muscle mass and fitness in adolescent and young adult survivors of hematopoietic cell transplantation (HemSCT) [127].

However, some preclinical studies showed that, compared to the CON group, chronic NRS (300 mg/kg/day for 21 days by oral gavage) decreases the exercise/swimming performance in Wistar rats. The redox features and pleiotropic metabolism of NAD^+^ and NADP^+^ could be an explanation for the potential inhibition of oral NR treatment on physical performance of rats [128,129,130].

NAD^+^ repletion improves MT activity and muscle stem cells (MuSCs) function and enhances life span in aged mice. In an experimental model of muscular dystrophy, NRS prevented the senescence of MuSC in the mdx (C57BL/10ScSn-Dmd(mdx)/J) mice [131,132]. Also, seeking treatment of MT myopathy, orally administered NR was found to induce MT unfolded protein response, MT biogenesis in brown adipose tissue (BAT) and skeletal muscle, prevent abnormalities of MT ultrastructure, and prevent deletion of mtDNA [133]. In this regard, NRS increased the levels of NAD^+^ in liver and skeletal muscle, exercise capacity, and MT respiration, alleviating the exercise intolerance in adenine nucleotide translocator 1 (ANT1)-deficient mice [134]. In the recent study on 20 body mass index (BMI)-discordant monozygotic twin pairs, NRS (250 mg to 1 g daily for five months) improved muscle MT biogenesis, satellite cell differentiation, and gut microbiota [135].

NAD^+^ boosting was preclinically characterized as a promising therapeutic approach for rheumatoid arthritis (RA) patients. NR and NMN treatments were ex vivo tested on RA–PBMCs, thus highlighting NAD^+^ amount increasing via NAMPT and nicotinamide mononucleotide adenylyl transferase (NMNAT), and reducing the proinflammatory, pro-oxidative and pro-apoptosis in RA patients [136].

The influence of NR (250 mM, 500 mM, 1 M) on the development and growth of *pectoralis major* muscle (PMM) was examined on fertilized Cobb 500 broiler eggs. NRS introduced by injection into the yolk and albumen influenced in ovo broiler myogenesis in a fashion correlated with an increase in PMM weight, length, and fiber density [137,138,139].

## 3. Nicotinamide Riboside as a Tool to Mitigate Metabolic Disorders

### 3.1. Nicotinamide Riboside and Obesity

Among NAD^+^ precursors, NR is often suggested as a dietary supplement to enhance oxidative metabolism and protect against HFD-induced obesity and deficient MT functionality for age-related diseases [25,140]. In aging mice, NAD^+^ supplementation via NR rejuvenated intestinal stem cells (ISCs) improved the repairing capacity of gut damages by increasing the activity of SIRT1 and mammalian target of rapamycin complex 1 (mTORC1) [141]. In experimental models of mildly obese male mice (C57BL/6N, C57BL/6J), NRS for eight weeks increased MT respiration. However, it had minimal impact on energy metabolism, without influencing body weight, internal organs weight, glucose metabolism, liver lipids amount, and metabolic adaptability [142].

NR- and NMN-conditioned microbiota reduced HFD-induced weight gain in C57BL/6J male mice by increasing their energy consumption. Here, butyrate-producing *Firmicutes* were enriched by FMT from NR-supplemented donors to HFD-fed naïve mice [143,144]. Alterations in intestinal brush border membrane functionality and bacterial populations following intra-amniotic administration (to *Gallus gallus*) were reported for 30 mg/mL doses of NR and its derivatives, water-soluble NR tributyrate chloride and oil-soluble NR trioleate chloride. *Bifidobacterium*, *Clostridium*, *Lactobacillus*, and *Escherichia coli* populations were significantly increased by NR treatment [145]. Further, oral supplementation of NR for 12 weeks alters gut microbial composition (*Erysipelotrichaceae*, *Lachnospiraceae* and *Ruminococcaceae* families) in HFD-fed rats and mice, but not in humans [146].

NRS exerts an anti-obesity effect and prevents inflammation and LF in white adipose tissue (WAT) of 8-week-old (young) and 16-week-old (old) female C57BL/6J DIO mice fed an HFD/high-sucrose diet/high-cholesterol diet or HFD combined with 400 mg/kg/day NR for 20 weeks. Weight and size of gonadal WAT (gWAT) adipocytes of old mice females were decreased by NRS [147]. Similar results were seen in recent research concerning NR and CR effects on gut microbiota and liver inflammatory and morphological markers in CAF DIO in adult male Wistar rats [148]. The effect of NR on lipid metabolism and gut microflora–bile acid axis was studied in alcohol-exposed mice. NRS increased the NAD^+^/NADH ratio, deoxycholic acid and hyocholic acid levels, and decreased the activation of the protein phosphatase 1 (PP1) signaling pathway, chenodeoxycholic acid, TG and total bile acid levels, and lipid accumulation [149].

Oral NRS also appeared to confer metabolic benefits in obese mice. Thus, NR induces a thermogenic response in the BAT of lean mice. Over five weeks, male C57BL/6J mice were supplemented with NR (400 mg/kg/day), giving a reduction in abdominal visceral fat depots and an increase in body temperature [150,151]. Also, compared with CONs, NRS conferred marginal metabolic benefits, augmenting MT functionality in C57BL/6NJ obese mice without remodeling the skeletal muscle acetyl-proteome [152]. NRS provided daily to suckling male mice improved lipid and energy metabolism in skeletal muscle and liver in adulthood, correlated with an upregulation of SIRT1 and AMP-dependent protein kinase signaling pathways [153]. In obese rats under CR, NRS (400 mg/kg for four weeks) neutralized hypothalamic inflammation by reverting high levels of TNF-α and increases weight loss without altering skeletal muscle mass [154]. Further, high doses (9 g/kg diet) of NRS for 18 weeks induced glucose intolerance and WAT dysfunction in male C57BL/6JRccHsd mice fed a mildly obesogenic diet (MOD) containing low but adequate tryptophan [155,156,157].

In an HFD mouse model, a reduction in obesity and insulin resistance was seen by dual targeting of visceral adipose tissue (VAT) and BAT using a novel combination of metabolic cofactors (NR, *N*-acetylcysteine, betaine, L-carnitine) orally administered for four weeks [158]. NRS (400 mg/kg daily) ameliorates high-fructose-induced lipid metabolism disorder in C57BL/6J mice via improving fibroblast growth factor 21 (FGF21) resistance in the liver and WAT. NR treatment upregulated the SIRT1–NF-κB pathway, decreasing inflammatory processes and increasing NAD^+^/NADH ratios [159]. Another study concluded that DNA methylation changes are associated with the programming of WAT browning features by RSV and NR neonatal supplementations in mice [160]. A positive signal was seen with a combined treatment of L-carnitine (0.4%, *w*/*w*) and NR (0.3%, *w*/*w*). This evidently improved hepatic metabolism and attenuated obesity and HS in HFD-fed Ldlr^−/−^ Leiden mice [161].

As a mechanism to treat metabolic disorders, NR may shift the differentiation of human primary white adipocytes to beige adipocytes. This may impact substrate preference and uncouple respiration through SIRT1 activation and MT-derived reactive species production [162]. In a randomized, double-blinded, PLA-controlled, crossover study in human volunteers (45–65 years, BMI 27–35 kg/m^2^), NR (1 g daily for six weeks) enhanced in vitro beta-adrenergic BAT activity and did not alter cold-induced thermogenesis [163]. A randomized, PLA-controlled, double-blind, clinical trial in 40 healthy obese (BMI > 30 kg/m^2^) insulin-resistant men looked at safety, insulin sensitivity, and lipid-mobilizing effects of NR (1000 mg twice daily) dietary supplementation for 12 weeks. It was found that NRS was safe and well tolerated, but did not improve either endogenous glucose generation, oxidation, or elimination, and did not improve insulin sensitivity [2]. Further, similar impacts of NRS were seen on endocrine pancreatic function and incretin hormones in non-diabetic men with obesity [164]. In a follow up study by the same authors, NRS does not alter MT respiration, content, or morphology in skeletal muscle in middle-aged, obese, and insulin-resistant men [165].

### 3.2. Nicotinamide Riboside and Diabetes

NRS can be used to mitigate type 2 diabetes (T2D) and neuropathy in HFD-fed prediabetic and diabetic male C57BL/6J mice. NR treatment significantly restored liver NADP^+^ and NADPH amounts and reduced weight gain, blood glucose and HS [166]. NAD^+^ precursors (NMN, NR) improved MT function in diabetes and prevented experimental diabetic peripheral neuropathy (DPN). The experiments were carried out in streptozotocin (STZ)-induced diabetic rats or mice. NMN was administered for two months (50 or 100 mg/kg i.p. injection on alternate days). HFD-fed mice were supplemented by NR at 150 or 300 mg/kg for two months. NR treatment in HFD-fed mice led to the normalization of adult dorsal root ganglion (DRG) neurons’ functionality [167]. From the point of view of cellular and molecular mechanisms, a recent study in NRK1 KO mice showed that NRK1 protected against HFD and age-induced pancreatic β-cell failure [168]. Also, NRS promoted mitofusin 2 (MFN2)-mediated MT fusion in diabetic hearts through the SIRT1–PGC-1α–peroxisome proliferator-activated receptor alpha (PPARα) pathway [169].

The effects appear to be broad. Supplementation with NR reduces brain inflammation and improves cognitive function in diabetic mice. NR treatment for six weeks of HFD-fed diabetic Institute of Cancer Research (ICR) male mice significantly reduced amyloidogenesis (amyloid beta (Aβ) precursor protein, presenilin 1) and neuroinflammation (IL-1, IL-6, and TNF-α markers) [10]. In STZ-induced diabetic rats, i.p. administered NR every 48 h for six weeks improved enteric neuropathy through myenteric plexus neuroprotection [170].

Other benefits of NRS in experimental murine diabetes refer mainly to the (i) alleviation of hepatic metaflammation by modulating NLRP3 inflammasome in a rodent (8-week-old KK/HlJ male mice) model of T2D [171], (ii) enhancement of endothelial progenitor cell (EPC) function to promote refractory healing of diabetic wounds through mediating the SIRT1–AMP-activated protein kinase (AMPK) pathway [172], and (iii) improvement in fetal growth under hypoglycemia previously induced in pregnant mice [173].

## 4. Nicotinamide Riboside for Healthy Aging and Longevity

### 4.1. Nicotinamide Riboside for Healthy Aging

NAD^+^ metabolism and homeostasis are important in aging and disease [174,175]. Declining NAD^+^ levels induce the augmentation of hypoxia-inducible factor-1alpha (HIF-1α) and a pseudohypoxic state disrupting PGC-1α/-1β-independent nuclear–MT communication during aging [176].

Thus, in recent preclinical studies, managing NAD^+^ deficits with NAM, NA, NR, and NMN provided healthy aging [177]. In the Caco-2 cell line and NIAAA mouse model, NAD^+^ supplementation by NR alleviated intestinal barrier injury induced by ethanol via protecting epithelial MT function [178]. NRS increased the formation of human leukocytes from hCD34+ progenitors in the immunodeficient mice model. Thus, through increased MT clearance, NR potently stimulated the hematopoiesis [179] and the lymphoid potential of Atm^−/−^ and old mice HSCs [180,181]. Moreover, NMN can be used to activate SIRT1 and improve the pathophysiology of diet- and age-induced diabetes in mice [182].

Likewise, reduced NMN is a newly analyzed and potent NAD^+^ precursor in mammalian cells and mice blood and internal organs [183]. In randomized, double-blind, PLA-controlled clinical trials in healthy middle-aged and older adults, chronic NRS (500 mg twice daily) is safe and well tolerated, and it is recommended for (i) increasing NAD^+^ level [8] and (ii) treating elevated systolic blood pressure (SBP) and arterial stiffness having initial above-normal (120–159 mmHg) SBP [184].

In aging and disease, a close connection relates NAD^+^ levels and the activation of SIRTs, as seen in murine models: (i) NR alleviates CPT-induced peripheral neuropathy (CIPN) and neuronal death via SIRT2 activation and enhancement of nucleotide excision repair in Lewis lung carcinoma model [185]; (ii) SIRT3 is required for regeneration of WT or mutant liver but not for the beneficial effect of NR [186]; (iii) SIRT3 deficiency aggravates contrast-induced acute kidney injury (CIAKI) in vitro (HK-2 cells) and in vivo (WT and SIRT3 KO mice) [187]; (iv) NR attenuates inflammation and OXS by activation of SIRT1 and normalization of NAD^+^/NADH ratio in alcohol-stimulated RAW 264.7 macrophages and in mouse bone marrow-derived macrophages [188].

NRS could be a missing piece in the puzzle of exercise therapy for older adults. Short-term oral NR treatment, 300 mg/kg or 600 mg/kg daily, improves muscle quality and function in middle-aged male C57BL/6J mice and increases cellular energetics and differentiating capacity of myogenic progenitors [189]. A randomized, PLA-controlled trial of safe and well-tolerated daily NRPT supplementation (1 g NR and 0.2 g PT) improved skeletal muscle regeneration after experimental muscle injury in 23 elderly individuals (55–80 years) [190]. Moreover, in a double-blind, crossover study, acute NRS improved redox homeostasis and exercise performance in 12 old individuals compared with the same number of young men [191].

Recent studies have highlighted that senotherapeutic NRS could contribute to healthy aging and longevity. As a sample of results: (i) the effects of senolytic drugs, including NR, were tested on human mesenchymal stromal cells (MSCs) with no significant action on molecular markers for replicative senescence [192]; (ii) the senotherapeutic NR triflate improved the NAD^+^ levels of buffy coat-derived platelet concentrates, but cannot prevent storage lesion for 23 days [193]; (iii) 17-α-estradiol late in life extends lifespan in aging UM-HET3 heterogeneous male mice, but NRS does not affect lifespan in either sex [194].

In this regard, NAD^+^ supplementation (NR, NAM, NA) exhibited emerging roles in replicative and chronological aging in fungi and mammals. For example: (i) *Saccharomyces cerevisiae YOR071C* gene encodes the high-affinity NR transporter Nrt1 polypeptide [195]; (ii) *S. cerevisiae* unicellular organism probably represents one of the most recognized experimental aging models for the study of replicative lifespan (RLS, proliferating cells) and chronological lifespan (CLS, non-proliferating cells) [196,197,198].

### 4.2. Brain Aging, Cognitive Impairment, and Neurodegenerative Diseases

Supplements of NAD^+^ precursors (NR, NRH, NMN) are a potential way to prevent cognitive decline within aging-associated diseases, such as neurodegenerative disorders [199,200,201]. For example, in models of murine dementia, NRS decreased neuroinflammation, DNA damage, and apoptosis while contributing to maintaining synaptic plasticity, integrity of the blood–brain barrier (BBB), and gut microbiota functionality. It also improved hippocampal synaptic plasticity, learning, and memory in AD. Aβ forming in the brain can be prevented by NRS partly through the upregulation of ubiquitination and proteasomal degradation of PGC-1α-mediated beta-secretase 1 (BACE1) [202,203].

These studies have extended for over a decade. For example, in Tg2576 mice, NR dietary supplementation (250 mg/kg/day) for three months significantly alleviated cognitive damage, increased the NAD^+^/NADH ratio in the cerebral cortex, and reduced Aβ production [26]. A further study examines DNA repair-deficient 3xTgAD/Polβ^+/−^ mice that exhibited cognitive impairment, synaptic dysfunction, phosphorylated *tau* (p-*tau*) pathologies, and neuronal death, the main characteristics of human AD. Here, NRS decreased neuroinflammation, DNA damage, and hippocampal neurons’ apoptosis, increased SIRT3 activity in the brain, improved cognitive function, and restored hippocampal synaptic plasticity [204]. NRS (2.5 g/kg in food for three months) in APP/PS1 transgenic AD and aged mice inhibited serum NAMPT elevation, astrocyte activation, neuroinflammation, senescence, Aβ accumulation, and astrocyte migration to Aβ, as well as improving locomotor activity, cognitive function, behavior, and dementia progression [205,206]. Inhibition of CD38 and NRS alleviated LPS-induced microglial and astrocytic neuroinflammation/neurodegeneration by increasing NAD^+^ levels in mice brains and by suppressing the NF-κB signaling pathway at the microglia level [207]. In APP/PS1 transgenic (AD) mice, NRS for eight weeks normalized gut dysbiosis for *Adlercreutzia*, *Akkermansia*, *Bacteroides*, *Bifidobacterium*, *Butyricicoccus*, *Desulfovibrio*, *Lactobacillus*, *Olsenella*, and *Oscillospira* microbiota species [208]. In late-onset AD patients, NR and caffeine co-administration partially restores diminished NAD^+^ availability but does not alter bioenergetic metabolism [209]. In a randomized, double-blinded, PLA-controlled, phase II clinical trial, combined metabolic activators (CMAs) administered in a single dose during the first 28 days and twice daily between days 28 and 84 significantly increased the cognitive capacity and alleviated NAD^+^ plasma levels and GSH metabolism of AD patients. CMA complex included 12.35 g L-serine (61.75%), 3.73 g L-carnitine tartrate (18.65%), 2.55 g *N*-acetyl-L-cysteine (12.75%), and 1 g NR (5%) [210].

NR food supplementation for 28 days rescues Ang II-induced cerebral small vessel disease (CSVD) in C57BL/6 mice. NRS significantly reduced glial activation, neuroinflammation, and white matter injury that is associated with cognitive dysfunction. It also supported BBB integrity and vascular remodeling, and improved Ang II-induced CSVD [211].

Further, treatment with NAD^+^ precursor NRS rescues MT defects in induced pluripotent stem cells (iPSCs) and aging-associated dopaminergic neuronal loss and motor decline and *Drosophila* models of glucocerebrosidase (GBA)-related Parkinson’s disease (PD) [212]. In a randomized, PLA-controlled, phase I clinical trial in 30 newly diagnosed, treatment-naïve PD patients (NADPARK study), oral NRS 1 g for 30 days significantly increased the level of NAD^+^ and its related metabolites in the CSF and decreased the inflammatory cytokines amounts also in serum and CSF [213]. In a recent randomized, PLA-controlled crossover trial in 22 healthy older adults using oral NRS (500 mg, twice daily, six weeks), the levels of NAD^+^ in neuronal-origin enriched plasma extracellular vesicles (NEVs) were increased, and levels of kinases (Aβ42, pJNK, pERK1/2) implicated in neuroinflammation and insulin resistance pathways were inhibited [214]. Moreover, in a double-blind, PLA-controlled trial of 29 PD patients, including cases with common pathogenic mutations in the methylenetetrahydrofolate reductase (*MTHFR*) gene, it was found that high-dose NRS for 30 days was not associated with altered DNA methylation homeostasis [215].

### 4.3. Aging and Cancer

NAD^+^ metabolism is a major feature of cancer pathogenesis (tumorigenesis), being closely related to genome integrity provided by efficient redox homeostasis, MT metabolism, and signal transduction [23]. The anti-tumoral effect of NR was studied in experimental models of hepatocellular carcinoma (HCC), such as the subcutaneous transplantation of tumors in Balb/c nude mice (xenograft) and C57BL/6J mice (allograft) and hematogenous metastatic tumor in nude mice. Daily administration of NR (400 mg/kg) by oral gavage extended the overall survival of HCC mice and decreased the size of allografted tumors and lung, liver, and bone metastases. Also, in vitro TGF-β-induced migration/invasion of HepG2 cells was inhibited by NRS [216].

In a mouse model of C26 adenocarcinoma, NR pellet dietary supplementation significantly improved cancer cachexia and inflammation through the inhibition of specific molecular markers, such as IL-6, TNF-α, PCG-1α, and muscle-specific ubiquitin-proteasome ligases (e.g., mitofusin-2, atrogin-1, muscle RING-finger protein-1 (MuRF-1)) [217]. Oral administration of 200 mg/kg NR in female tumor-bearing rats in a preclinical model of mammary gland cancer induced by *N*-methyl-nitrosourea (MNU) and previously treated with PTX i.v. injections (three doses of 6.6 mg/kg) led to a decrease in tumor growth and the Ki67 index of tumoral cells and to an improvement in peripheral neuropathy symptoms [218]. In high-risk skin cancer patients, boosting NAD^+^ with NR, MNM, and NAM p.o. supplements decreased the incidence of keratinocyte carcinoma. This presumably occurred through its cellular protective effects, mainly targeting DNA repair and prevention of possible activation of oncogenic mutations [219].

## 5. Safety and Bioavailability of Nicotinamide Riboside

### 5.1. Nicotinamide Riboside Safety

The safety of NRCl was assessed by biological assays, including bacterial reverse mutagenesis tests (Ames test), in vitro chromosome aberration tests, in vivo micronucleus tests, and toxicological studies in rats. These covered acute (24 h), sub-acute (14-day), and chronic (90-day) administration. Genotoxicity of NRCl was not observed, and lethality was absent even at a dose of 5000 mg/kg (p.o.). NRCl had the lowest-observed-adverse-effect level (LOAEL) and the no-observed-adverse-effect level (NOAEL) at the dose of 1000 mg/kg/day and 300 mg/kg/day, respectively. Here, the liver, kidney, ovary, and testicles were the target organs for oral toxicity assays [220].

A randomized, double-blind, PLA-controlled clinical trial for eight weeks was performed in overweight healthy women and men who received different doses of NRCl (orally 100 mg, 300 mg, 1000 mg). Between NRCl and PLA groups or between different NRCl groups, neither important differences nor adverse effects were observed. However, within two weeks of NRCl supplementation, significant increases in serum levels of NAD^+^ and NAD^+^ metabolites were recorded [3].

In a randomized, double-blind, PLA-controlled study, the safety and efficacy of NRPT (Basis™) were assessed for eight weeks in 120 healthy adult volunteers (60–80 years old). In addition to the significant increase in NAD^+^ blood levels, repeated doses of NRPT supplementation did not cause any adverse effects [221]. The European Food Safety Authority (EFSA) Panel on Nutrition, Novel Foods and Allergens (NDA) of the European Commission (EC) concluded that NR and NRCl are “safe for the healthy adult population, excluding pregnant and lactating women, and that an intake of up to 230 mg/day is safe for pregnant and lactating women”. Moreover, for the two active compounds, there are no concerns about genotoxicity and their safety/tolerability in human studies [222,223,224]. Three more leading international regulatory authorities, including the U.S. Food and Drug Administration (FDA), Health Canada (HC), and the Therapeutic Goods Administration (TGA) of Australia, concluded that NR is safer for human use than other NAD^+^ metabolites [225].

Similar studies were performed on NR-related compounds. A non-randomized study of single oral supplementation within the daily tolerable upper levels of NAM affects blood NAM and NAD^+^ amounts in six healthy humans. NAM daily intake of up to 200 mg was found, at 0.5 h, a 30-fold increase in its maximum amount in the whole blood of male volunteers. At 12 h, NAD^+^ serum concentrations have reached the maximum level. Metabolomic variations due to NAM dietary supplementation lasted one day, and then after two days, they returned to the baseline [226]. Moreover, in another study, oral supplementation of NAM (500 mg) significantly increased blood NAD^+^ levels after 12 h in a cohort of five healthy adult subjects [227].

A toxicological study assessed the safety of Restorin^®^ NMN, a direct NAD^+^ precursor (β-NMN) in a high-purity formulation, by oral administration at 500, 1000, and 2000 mg/kg/day in Sprague–Dawley rats over three months followed by a recovery period of two weeks. No adverse effects were observed at the dose of 500 mg/kg/day. Restorin^®^ NMN has a LOAEL of 2000 mg/kg/day and a NOAEL of 500 mg/kg/day in male rats and 1000 mg/kg/day in female rats [228].

In a double-blind, PLA-controlled, randomized clinical trial, oral administration of NMN (250 mg/day for 12 weeks) proved to be safe by increasing the level of blood NAD^+^ in 30 healthy volunteers. A significant increase in the level of NA mononucleotide (NAMN), rather than in the level of NMN, was also seen [229]. MIB-626, an oral formulation of a microcrystalline unique polymorph of β-NMN, increased circulating NAD^+^ levels in a double-blind, PLA-controlled trial in 32 overweight or obese middle-aged and older adults (55–80 years). Once or twice daily, 1000 mg MIB-626 supplementation proved to be safe and well tolerated. Age, sex, or BMI did not influence the variations in serum amounts of NMN or NAD^+^ [230].

### 5.2. Nicotinamide Riboside Bioavailability

In mice and humans, NR is the most orally bioavailable of NAD^+^ precursors. Amounts of NAD^+^ in mouse liver were elevated by oral NRS with better pharmacokinetics (PK) than seen with NA and NAM. In one healthy volunteer, a pilot study showed that a single dose of oral NRS elevated serum NAD^+^ levels as much as 2.7-fold. The first clinical trial of NR PK in healthy humans highlighted that 100, 300, and 1000 mg of NR single dose increased the NAD^+^ metabolome in a dose-dependent manner [231].

An open-label, non-randomized study of the PK of the NR nutritional supplement and its effects on blood NAD^+^ levels was performed in eight healthy volunteers. Oral NR (250 mg) was administered daily for the first two days. Then, the maximum dose was increased to 1000 mg twice daily on days 7 and 8. NRS did not induce adverse effects and was well tolerated. In NR and NAD^+^ levels, starting from baseline to the ninth day, the absolute changes were highly correlated [232,233].

BM stromal cell antigen 1 (BST1) regulates NR metabolism via its dual roles in glycohydrolase and base-exchange enzyme activities. Two different pathways contribute to the increase in NAD^+^ level after oral NR treatment: (i) NR salvage pathway in the early phase of NR direct absorption, and (ii) Preiss–Handler pathway in the late phase of NR hydrolysis to NAM by BST1 followed by transformation to NA by the distal gut microbiome [234,235].

Pre-steady-state and steady-state kinetic analyses were made for the *N*-ribosyl hydrolase activity of hCD157. Cell-surface hCD157 binds and catalyzes the slow hydrolysis of NR and NA riboside (NAR) after the formation of an initial complex with NAD^+^ derivatives [236].

The import of NR and NAR into human cells is mediated by the equilibrative nucleoside transporter (ENT) family of proteins, ENT1, ENT2, and ENT4. After entering the HEK293 cells, NR is activated by NRK phosphorylation in the cytosol and then transformed to yield NAM [237]. NMN and NR metabolism in mammalian cells is predominantly controlled by NRK1 and purine nucleoside phosphorylase (PNP). Simultaneous PNP downregulation could improve the health advantages of NRS [14,238].

Kulikova et al. (2015) studied the generation, release, and uptake of the NAD^+^ precursor NAR by human cells. Under normal culture conditions, untransfected HeLa cells generated and released a sufficient level of NAR and NR as whole compounds derived from NAD^+^ metabolism [239].

A pilot study investigated changes in the human plasma and urine NAD^+^ metabolome during a 6 h i.v. infusion of 3 μM/min NAD^+^. No changes were seen after two hours in the levels of plasma NAD^+^ or its metabolites, such as NAM, MeNAM, adenosine diphosphate ribose (ADPR), and NMN. After six hours, increased urinary excretion of NAD^+^ and MeNAM was seen; at the same time, NAM urinary level was not increased [240].

In a simulated intestinal fluid using porcine pancreatin, NR trioleate chloride, a newly synthesized hydrophobic NR derivative, was partly digestible and easily released NR [241].

NRH is a potent enhancer of NAD^+^ levels in vitro in mammalian cells, significantly increasing NAD^+^/NADH ratio, as well as in vivo after i.p. injection to C57BL/6J mice [242]. Although there are minor structural dissimilarities, NRH defines a new pathway for NAD^+^ biosynthesis, independent of NRK1, and acts as an orally bioavailable NAD^+^ precursor also in mammalian cells and mouse tissues [243]. In a recent study, it was found that NR and dihydronicotinic acid riboside synergistically act to increase NAD^+^ intracellular levels by generating NRH [244].

## 6. Borate-Stabilized Nicotinamide Riboside

This collection of research results is cited to encourage the consumption of NR as a dietary supplement. Indeed, several companies now offer NR in pill form for this purpose. Unfortunately, NR is a reactive species, often unstable during manufacture, transport, and storage. Accordingly, HPLC analysis of many of the commercially available products found them to not be pure NR but rather mixtures of NR and various other materials. Often, the NR is present in small amounts. These facts led us to develop NR borate and less-expensive routes to make it. These are reviewed below.

### 6.1. Prebiotic Synthesis

While this is irrelevant to the use of NR as a dietary supplement, it is of conceptual interest to note that the synthesis of NR borate was grounded in considerations of prebiotic chemistry. The analysis began with the observation that the chemical and biological syntheses of NR are extremely challenging, with the challenge centered on the bond that joins the NAM ring to the ribose ring. Thus, while several methods of chemical and biological syntheses of NR are reported [245], these are rather expensive. At the time of this writing, only one U.S. company (ChromaDex) produced NRCl [3,220]. Further, NR is unstable. When analyzed by HPLC and other standard analytical methods, many samples of commercial NR are, in fact, mixtures of products that indicate the decomposition of the material [246].

Prebiotic chemistry was applied to this problem. Here, Kim and Benner discovered that NR phosphate emerges in stable form by direct reaction of ribose-1,2-cyclic phosphate with NAM [247,248]. Ribose-1,2-cyclic phosphate is available inexpensively from ribose and amidotriphosphate, which is available from the very inexpensive cyclic trimetaphosphate and ammonia. The phosphorylated NR product is then enzymatically dephosphorylated by a phosphatase [249]. In the final stage, NR is stabilized by borate, resulting in NR borate [250]. Borate-stabilized NR was found to have high stability against thermal and chemical degradation [250,251]. This mitigates problems in current commercial NR.

The borate might be seen as simply a stabilizing species. However, recent research suggests that borate itself might be a key element in NAD^+^ metabolism [252,253]. Borate released by the ribose–borate complex by hydrolysis could be a reservoir of physiological boron (B). In addition, borate esters can remain undigested in the upper gastric system and thus be more accessible to the microbiota [254]. According to new insights into the essentiality of B in the healthy symbiosis between the microbiota and the human host, NR borate becomes an interesting prebiotic that delivers B that is essential to bacteria for the synthesis of autoinducer-2 (AI-2) and for strengthening the colonic mucus [254,255,256].

The prebiotic synthesis and borate stabilization of NR had two very important advantages: (*i*) it is more cost-effective due to less expensive starting materials; (*ii*) NR is more stable, mitigating the fact that many samples of commercial NRCl are mixtures of products that indicate decomposition of the material (Figure 1).

### 6.2. Solubility and Degradation Kinetics of Nicotinamide Riboside Borate

The possibility that NR borate provides, at last, a stable form of NR should motivate the study of its effect in preclinical and clinical studies. Since borate esters are dynamic, with borate dissociating from NR upon dilution, we expect that the effects of NR borate should closely parallel the effects of NR.

The solubility and degradation kinetics of NR borate were examined in various physiological media, including 0.1 N hydrochloric acid (pH 1.5), sodium acetate buffer (pH 5.0), water (pH 7.0), and phosphate-buffered saline (pH 7.4), using HPLC analysis. Furthermore, stability studies were conducted to investigate how different pH levels and temperatures influenced the degradation kinetics of NR borate. The results revealed significant variations in the solubility and stability of NR borate within the tested biological solutions. The solubility of NR borate was found to be pH-dependent, increasing as the pH level rose [257].

Using the shake-flask method at room temperature, the solubility of NR borate was evaluated. The measurements showed that NR borate had solubilities of 1972.7 ± 15.4 mg/mL, 1060.5 ± 31.0 mg/mL, and 926.0 ± 34.4 mg/mL at pH 1.5, 5.0, and 7.4, respectively. These findings indicate that NR borate demonstrated high solubility in all three solutions with varying pH levels, meeting the standards outlined in the U.S. Pharmacopoeia recommendations [257] (Figure 2).

To investigate the spontaneity of NR borate degradation in various pH solutions, the Gibbs free energy (Δ*G*) was calculated. The Δ*G* value obtained for NR borate was 2.43 kcal/mol [257].

When examining NR borate as a nutritional supplement, it is important to investigate the generation of NAM and the ribose–borate complex, which acts as a physiological reservoir of B, as degradation byproducts. It is understood that the existence of anionic borate can impede the conversion of NR into NAM and trigger the activation of SIRT. While there is limited scientific data on the slight degradation of NR in simulated gastric fluids, substantial degradation seems to occur in the intestine due to its elevated pH levels [233]. The stability of NR borate is significantly influenced by neutral conditions, exemplified by the pH 7.4 employed in this study. In the Arrhenius model of NR borate degradation, the rate of degradation is higher at pH 7.4 compared to pH 1.5 and pH 5.0, supporting the notion that the degradation of NR borate is influenced by the concentration of HO^−^ ions in the solution. It is crucial to exercise caution when using an aqueous solution of NR borate, particularly at high pH values where an excess of HO^−^ ions can accelerate the degradation process due to hydrolysis. Our findings suggest that there may be a slightly distinct mechanism of degradation for NR and NR borate under acidic conditions compared to neutral or alkaline conditions. Consequently, this model can serve as a general reference for comprehending the stability of NR borate in various buffered solutions. Additionally, temperature plays a role in the content of NR borate in solution, with a 10 °C increase approximately doubling its degradation rate under any pH condition.

The estimation of *log*P is crucial for predicting the permeability of NR borate. It is well established that compounds with estimated *log*P values less than 1.72 exhibit low permeability. According to the Biopharmaceutical Classification System (BCS), NR borate is classified as Class III due to its high solubility but anticipated low permeability, as indicated by a *log*P value of −4.17. This physiological characteristic may be linked to the proportion of released degradation products. Ultra-high-performance liquid chromatography (UHPLC) spectra revealed that the degradation products of NR borate in aqueous solutions are NAM and the ribose–borate complex. Consequently, the concentration of NAM can serve as a marker for the degradation of both NR borate and NRCl, the commercially available form. In this study, a developed high-performance thin-layer chromatography (HPTLC) method was employed to separate and quantify NR and NAM peaks with a remarkable separation resolution (*R* = 1.8), surpassing the acceptable resolution (*R* > 1) recommended by the U.S. FDA [257]. This method offers a rapid means of detecting NR, its degradation product NAM, and the ribose–borate complex. Our research data present, for the first time, a definitive upper-temperature limit for the processing of NR borate and its products, which can impact the production of supplements at various stages. The optimization of NR borate administration will have a significant influence on its efficacy. Our study aimed to uncover the mechanism by which NR borate degradation produces a physiological product when exposed to simulated physiological fluids in vitro. The development of an industrial pilot NR borate opens up possibilities for further research, which could have a profound impact on its potential therapeutic applications. Consequently, under these conditions, NR borate solutions and formulations have the potential for high stability. The findings of this study will prove valuable for pharmaceutical scientists involved in manipulating NR borate working solutions and designing appropriate formulations for NR borate delivery. We observed a clear dependence of NR borate solubility on pH. NR borate has exhibited extreme instability at pH levels greater than 8.0 while remaining stable in both acidic and neutral environments.

As a result, NR borate may offer advantages over NRCl as a nutritional supplement, primarily due to the boric acid (BA) residue’s ability to block the glycosidic bond between the pyrimidine base and ribose. The study has shed light on the key physicochemical properties that should be considered when utilizing NR borate in both in vitro and in vivo experiments. The described HPTLC method has the potential for distinguishing between NR borate and its degradation products in environments with pH conditions similar to those found in the GI tract. The obtained data also hold significance for the processing, production, and storage of this supplement.

In addition, the greatest direction of the use of NR borate could be as a prebiotic compound. The latest research shows that the presence of NR in the colon has a very high nutritional value, and through its ester form of BA, it can become a nutraceutical with a role in healthy symbiosis as an essential B element as well as an essential nutrient in the health metabolism of the human microbiome [143,144]. In vivo, B organic compounds have been shown to be less toxic than BA and inorganic borates [253].

## 7. Conclusions and Perspectives

NAD^+^ is known classically as a metabolite that stands astride both catabolic and anabolic pathways throughout the metabolism that is taught in introductory biochemistry courses. However, non-classical studies starting over a decade ago found that it is also involved in higher-order functions, in part because of its involvement in the activation of SIRTs and the support of the MT unfolded protein response. Many studies have suggested that NAD^+^ is involved in an extensive spectrum of pathologies, including neurodegenerative disorders, cardiomyopathy, obesity, and diabetes. Further, healthy aging and longevity appear to be closely related to NAD^+^ and its related metabolites, including through NR and NMN. This system appears to have prophylactic and therapeutic value in improving age-associated neurodegenerative, CV, and metabolic diseases and conditions.

Accordingly, many are now recommending the consumption of materials in this system as dietary supplements, hoping to achieve overall improvements in human health. Among NAD^+^ precursors, NR appears to have special values. These include better tolerance, better uptake, and overall greater potency.

Unfortunately, NR is a reactive molecule, often unstable during its manufacturing, transport, and storage. Indeed, HPLC analyses of many commercial samples of NR show that they contain substantial amounts of material that are not, in fact, NR. More stable derivatives of NR that are easily converted upon consumption into NR are therefore desired.

Recently work related to prebiotic chemistry provided the borate derivative of NR. NR borate is considerably more stable than NR itself. However, immediately upon consumption, the borate dissociates from the NR borate and is lost in the body through dilution and binding to other species, notably carbohydrates such as fructose and glucose. The NR left behind is expected to behave pharmacologically in ways identical to NR itself.

This review provides a comprehensive uncritical summary through Q1 of 2023 of the literature that makes the case for the consumption of NR. It then summarizes the challenges of delivering quality NR to consumers using standard synthesis, manufacture, shipping, and storage approaches. It concludes by outlining the advantages of NR borate in these processes.

## Figures and Tables

**Figure 1 molecules-28-06078-f001:**
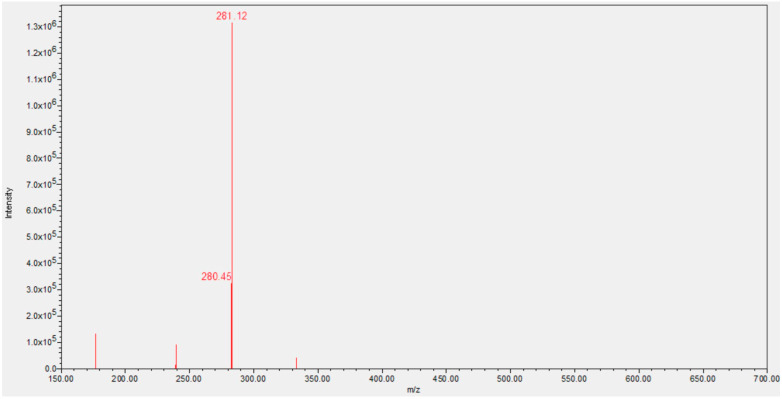
NR borate mass spectrum: ^11^B (*m*/*z* 281); ^10^B (*m*/*z* 280). B: boron; NR: nicotinamide riboside.

**Figure 2 molecules-28-06078-f002:**
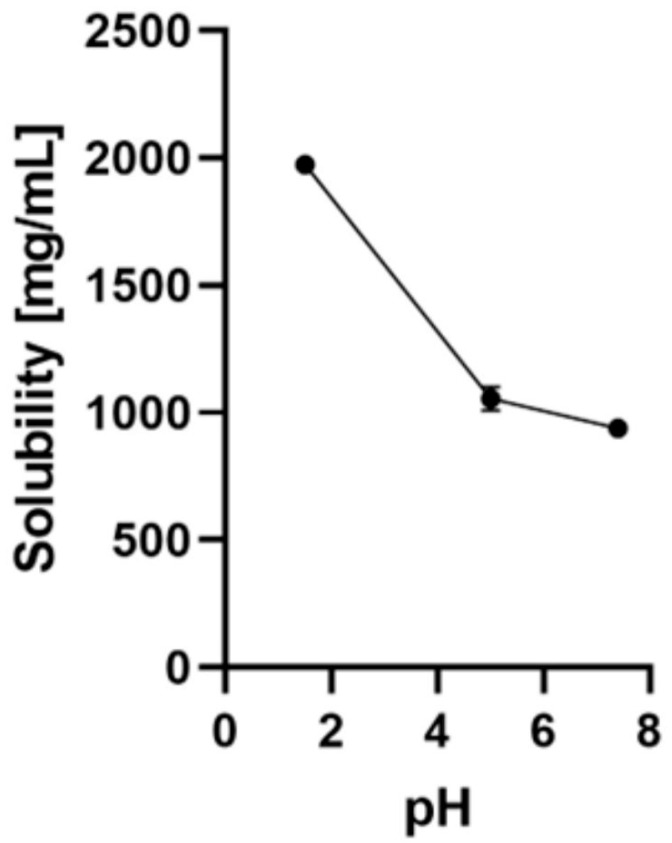
pH stability for NR borate.

## Data Availability

Data described in the manuscript will be made publicly and freely available without restriction at: https://drive.google.com/drive/folders/1BraUbHcRZ4vNJfuEIAzlzwsd-o4BSxMX?usp=sharing (accessed on 20 June 2023).

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
