# Peer review of "Nicotinamide Riboside, a Promising Vitamin B3 Derivative for Healthy Aging and Longevity: Current Research and Perspectives"

_molecules, 2023, doi:10.3390/molecules28166078_

Round 1

Reviewer 1 Report

comments and suggestions:

1.The topic of this review is about the effect of nicotinamide riboside on healthy aging and longevity. However, only part 4 is about this issue. A better title which can cover all the contents is needed.

2. Even though a lot of literatures were cited, a better summary of the literatures is needed, beyong just copying the conclusions of some of them.

3. Some paragraph just describes one literature, which should combine with other paragraphs.

4. In this review, nicotinamide riboside is abbreviated as NaR, which is usually as NR in other literatures. Nicotinic acid as NaC, which is usually NA. While nicotinamide is abbreviated as NA, which is usually as NAM in others. Please use the commonly used abbreviation.

5. It is better not to abbreviate borate as "B"  in the review. It is really confusing in the sentences.

Author Response

Nicotinamide Riboside, a Promising Vitamin B3 Derivative for Healthy Aging and Longevity: Current Research and Perspectives

Manuscript ID: molecules-2499727

Dear Reviewer,

First of all, we would like to address you many thanks for your accurate observations and valuable comments. We used all these and improved the paper accordingly.

All changes in the revised manuscript were marked up using the “Track Changes” function.

The following changes have been made for the Manuscript (ID molecules-2499727):

Reviewer’ questions/comments:

  1. The topic of this review is about the effect of nicotinamide riboside on healthy aging and longevity. However, only part 4 is about this issue. A better title which can cover all the contents is needed.

Answer:

The effects of nicotinamide riboside (NR) on healthy aging and longevity are closely related to the effects of NR on several organs and systems, and to the use of NR as a tool to mitigate metabolic disorders.

  1. Even though a lot of literatures were cited, a better summary of the literatures is needed, beyond just copying the conclusions of some of them.

Answer:

Our review provides a comprehensive uncritical summary through Q1 of 2023 of literature that makes the case for the consumption of NR. It then summarizes the challenges of delivering quality NR to consumers using standard synthesis, manufacture, shipping, and storage approaches. It concludes by outlining the advantages of NR borate in these processes.

  1. Some paragraphs just describe one literature, which should combine with other paragraphs.

Answer:

For a unified aspect of the presentation, some paragraphs have been combined with other paragraphs (See lines 91-96, 97-103, 104-109, 559-576, 1302-1308, 1310-1319, 1320-1489).

  1. In this review, nicotinamide riboside is abbreviated as NaR, which is usually as NR in other literatures. Nicotinic acid as NaC, which is usually NA. While nicotinamide is abbreviated as NA, which is usually as NAM in others. Please use the commonly used abbreviation.

Answer:

The three abbreviations have been modified in the manuscript, as follows: “NR” instead of “NAR”, “NA” instead of “NAC”, and “NAM” instead of “NA”.

  1. It is better not to abbreviate borate as “B” in the review. It is really confusing in the sentences.

Answer:

“NARB” abbreviation has been removed from the manuscript and it has been replaced with the “NR borate” term.

We have also introduced other additions/modifications that we hope will improve the quality of the manuscript:

â–ª The institutional e-mail address for corresponding author has been modified: “romulus.scorei@naturalresearch.ro” instead of “romulus_ion@yahoo.com” (See line 29).

â–ª Other abbreviations were also modified in the manuscript: “NRCl” instead of “NARC”; “NRS” instead of “NARS”; “NRPT” instead of “NARPT”; “NRH” instead of “NARH”; “NAMN” instead of “NAcMN”; “NAR” instead of “NAcR”; “MeNAM” instead of “MeNA”; “NARTBC” and “NARTOC” abbreviations have been removed from the manuscript.

â–ª Some grammar, stylistic or spelling errors have been corrected.

Kind regards,

Ion Romulus SCOREI, Professor, MD, PhD

Reviewer 2 Report

The authors have stated that they have set out to review nicotinamide riboside and its role in healthy aging and longevity.

This review appears reasonably comprehensive. However the authors discussion of their own area of interest in nicotinamide riboside borate should include a summary of potential boron/borate toxicity.

In addition there were occasional mistakes of fact throughout the article, including:

Line 49- NR is not generally referred to as an advanced form of vitamin B3. This statement may be the authors opinion but should not be stated as scientific fact.

Line 84/85 Nicotinamide does NOT cause flushing

Line 915 One molecule of NRborate can not produce one molecule of NMN and one molecule of ribose borate...this would require 2 molecules of ribose in the original NR-borate

There were quite a few examples of poor English grammar throughout the manuscript. A revised manuscript should be edited by a competent English grammar expert to improve readability

Round 2

Reviewer 1 Report

The questions are well addressed.